# Enhancing the Decolorization of Methylene Blue Using a Low-Cost Super-Absorbent Aided by Response Surface Methodology

**DOI:** 10.3390/molecules26154430

**Published:** 2021-07-22

**Authors:** Nor Hakimin Abdullah, Mazlan Mohamed, Norshahidatul Akmar Mohd Shohaimi, Azwan Mat Lazim, Ahmad Zamani Abdul Halim, Nurasmat Mohd Shukri, Mohammad Khairul Azhar Abdul Razab

**Affiliations:** 1Advanced Materials Research Cluster (AMRC), Faculty of Bioengineering and Technology, Universiti Malaysia Kelantan (UMK), Jeli 17600, Kelantan, Malaysia; mazlan.m@umk.edu.my; 2Faculty of Applied Sciences, Universiti Teknologi MARA Pahang, Bandar Tun Razak 26400, Pahang, Malaysia; akmarshohaimi@uitm.edu.my; 3Department of Chemical Sciences, Faculty of Science and Technology, Universiti Kebangsaan Malaysia, Bangi 43600, Selangor, Malaysia; azwanlazim@ukm.edu.my; 4Faculty of Industrial Sciences and Technology, College of Computing and Applied Science, Universiti Malaysia Pahang, Gambang, Kuantan 26300, Pahang, Malaysia; ahmadzamani@ump.edu.my; 5School of Health Sciences, Universiti Sains Malaysia, Kubang Kerian 16150, Kelantan, Malaysia; nurasmatms@usm.my

**Keywords:** activated carbon, agriculture waste, methylene blue, optimization, wastewater treatment

## Abstract

The presence of organic dyes from industrial wastewater can cause pollution and exacerbate environmental problems; therefore, in the present work, activated carbon was synthesized from locally available oil palm trunk (OPT) biomass as a low-cost adsorbent to remove synthetic dye from aqueous media. The physical properties of the synthesized oil palm trunk activated carbon (OPTAC) were analyzed by SEM, FTIR-ATR, and XRD. The concurrent effects of the process variables (adsorbent dosage (g), methylene blue (MB) concentration (mg/L), and contact time (h)) on the MB removal percentage from aqueous solution were studied using a three-factor three-level Box–Behnken design (BBD) of response surface methodology (RSM), followed by the optimization of MB adsorption using OPTAC as the adsorbent. Based on the results of the analysis of variance (ANOVA) for the three parameters considered, adsorbent dosage (*X_1_*) is the most crucial parameter, with an *F*-value of 1857.43, followed by MB concentration (*X_2_*) and contact time (*X_3_*) with the *F*-values of 95.60 and 29.48, respectively. Furthermore, the highest MB removal efficiency of 97.9% was achieved at the optimum *X_1_*, *X_2_*, and *X_3_* of 1.5 g, 200 mg/L, and 2 h, respectively.

## 1. Introduction

Over one third quarter of the Earth’s surface consists of water. Today, rivers and oceans face environmental issues related to water pollution and contamination, with textile industries as the primary source of these issues. More than 8000 chemicals used in the textile industry are harmful and poisonous to human health and aquatic ecological systems [1]. Even worse, the environmental pollution of rivers and soils by dyes could directly interfere with the photosynthesis process, leading to increased toxicity of living beings [2]. Moreover, the environmental problems could worsen due to high levels of heavy metals in dyes (e.g., zinc, lead, copper, mercury, and nickel) [3]. Thus, several reliable methods have been studied for colored wastewater treatment, namely flocculation [4,5,6], coagulation [7,8,9], membrane separation [10,11], bacterial strains [4], and adsorption [12,13,14].

Adsorption involves the adhesion of substances present in a liquid phase onto a solid surface [15]. Colored water is commonly treated using adsorption, which offers several advantages, being a straightforward and low-cost process, as well as having a high efficiency and versatility [16,17,18]. The adsorption and removal of dyes from wastewater is usually performed using activated carbon (AC) due to various benefits of employing AC containing a high carbon content, including simple operation, high porosity, and adsorption capacity [19,20]. AC can be prepared from agriculture biomass, such as coconut shell [21,22], corncob [23], durian shell [24,25], and oil palm [26,27]. Many solid wastes from different parts of the oil palm, including trunks, fibers, and empty fruit bunches, are produced in Malaysia, and these parts are good sources for producing AC [28]. Oil palm trunk (OPT) consists of 22.6% lignin, 21.2% hemicellulose, 39.9% cellulose, 1.9% ash, 3.1% wax, and 11.3% other constituents, which would be beneficial as raw materials to prepare AC as an adsorbent [29]. The conversion of OPT as an agricultural waste into a new, beneficial product would reduce environmental impacts, and this approach is in line with the waste-to-wealth concept.

A three-level factorial design is needed for the Box–Behnken design (BBD) of response surface methodology (RSM) when conducting an experiment. BBD–RSM can be implemented to obtain optimum adsorption conditions with the minimum number of experiments and least amount of chemicals. This approach can also be applied for the investigation of the effect of independent variables (factors) on the process and their influence on a dependent variable (response) [30]. BBD–RSM has been used to optimize adsorbate removal using AC prepared from agriculture waste [31,32,33,34,35,36], zeolite [37], and biopolymers [38]. Nevertheless, to the best of the authors’ knowledge, no study has been conducted for the optimization of synthetic dye removal using AC prepared from OPT.

In this study, OPT was used as a raw material for the preparation of oil palm trunk activated carbon (OPTAC), which was then characterized using scanning electron microscopy (SEM), Fourier transform infrared-attenuated total reflection (FTIR-ATR), and X-ray diffraction (XRD). Methylene blue (MB) removal was conducted to evaluate the adsorption capacity of OPTAC. Ultraviolet-visible (UV-vis) spectroscopy was applied to optimize the MB removal, based on a full-factorial experimental design. The effect of independent variables (adsorbent dosage, MB concentration, and contact time) on the MB removal percentage from aqueous solution was studied by utilizing OPTAC as the adsorbent. Moreover, a BBD design was utilized to determine the optimal MB removal and investigate the relationship between independent variables and MB removal.

## 2. Results

### 2.1. Characterization of OPTAC

#### 2.1.1. Scanning Electron Microscopy

Figure 1 presents the surface morphology of the native and activated carbon produced using OPT. No distinct pores could be observed on the surface of the OPT before chemical activation (Figure 1a). Meanwhile, good pore formation could be observed on the prepared OPTAC of different shapes and sizes (Figure 1b–d). Chemical activation eased the enlargement of pores in the carbon structure [39] and the pore development on the AC surface due to the evaporation of H_3_PO_4_ during carbonization [40]. The most noticeable changes that occurred after the activation were cracks and basic pores due to the volatilization of cellulose, hemicellulose, lignin content, and moisture of the carbon [41]. The successful activation process and the porous structure of the OPTAC will improve the efficiency of the adsorption process.

#### 2.1.2. FTIR-ATR Spectra of OPTAC

The FTIR-ATR spectra of the native and activated carbon of OPT are presented in Figure 2, where significant changes can be seen by comparing the spectra of native carbon (NC) (Figure 2a) and OPTAC (Figure 2b). The spectrum of NC of OPT consists of several significant peaks at 3353.3 cm^−1^, suggesting the characteristic absorption peak of O–H and N–H symmetric stretching vibration, 2111.5 cm^−1^ for C≡C stretching vibration of alkynes, 1574.2 cm^−1^ denoting C=O and C=C stretching vibrations, 1372.6 cm^−1^ (carboxylic acid O–H), 1119 cm^−1^ for C–H in-plane deformation, and 749 cm^−1^ showing C–H symmetric stretching vibration [42]. There are slight additions and changes in the FTIR-ATR spectrum for OPTAC compared to NC due to the acidic activation process, which modified its surface properties. The chemical treatment of OPTAC altered its carboxylate group, which explains the disappearance of the peak at 1372 cm^−1^ for the spectrum of OPTAC, whereas the appearance of peaks at 1796.2 cm^−1^ (stretching vibration of C=O in –COOH), 1557.5 (cm^−1^ assignable to stretching vibration of C=C), and 953 cm^−1^ correspond to the stretching vibration of C–O, C–X, or C–C [43]. The OPTAC spectrum has fewer strong peaks due to the heating effect during the activation process, where the decomposition of lignin, cellulose, and hemicellulose took place [44].

#### 2.1.3. XRD of OPTAC

Figure 3 exhibits the XRD results for native carbon (NC) and OPTAC. The weak and broad patterns near 2θ = 24° and 2θ = 43° in the spectra are due to the amorphous phase of the samples [45]. In addition, broad diffraction was observed in all samples near 2θ = 24° due to the microcrystalline graphite multilayers [46].

### 2.2. Response Surface Methodology–Box–Behnken Design

Table 1 presents the experimental results obtained by studying MB removal optimization using the BBD, where 17 experimental runs were performed randomly. Three independent variables (adsorbent dosage, MB concentration, and contact time) were selected to determine the most influential factors for MB removal. The experimental results based on the full-factorial BBD were fitted to the quadratic equation using multiple regression analysis. The quadratic model for MB removal (response) as a function of three independent variables is shown in Equation (1): MB Removal (Y) = 94.86 + 18.78 ∗ *X_1_* − 4.26 ∗ *X_2_* + 2.37 ∗ *X_3_* + 1.54 ∗ *X_1_* ∗ *X_2_* − 0.39 ∗ *X_1_* ∗ *X_3_* + 2.09 ∗ *X_2_* ∗ *X_3_* − 11.50 ∗ *X_1_*^2^ − 7.23 ∗ *X_2_*^2^ − 7.43 ∗ *X_3_*^2^(1)
where *Y* is the estimated response (MB removal), and *X_1_*, *X_2_*, and *X_3_* are the independent variables of adsorbent dosage, MB concentration, and contact time, respectively.

Table 2 presents the ANOVA results based on the developed quadratic model for MB removal under the designated experimental conditions. From the results, the model is significant, with the model *F*-value of 304.23 and *p*-value of 0.001. In addition, noise contributed to 0.01% probability of the model *F*-value. According to the findings, *X_1_* was the most influential parameter, with an *F*-value of 1,857.43, followed by *X_2_*, with an *F*-value of 95.60. The least influential parameter was *X_3_,* with an *F*-value of 29.48. The model is significant if the *p*-value is less than 0.05. The obtained lack-of-fit value of 1.82 proved that the lack-of-fit is not important compared to the pure error. The lack of fit had a 36.36 percent chance of occurring due to noise. The model was good and fit well based on the insignificant lack-of-fit. Furthermore, the response was significantly influenced by the interaction between *X_1_X_2_* and *X_2_X*_3_, according to the significance of the model (i.e., *p*-value) set at 0.05.

The coefficient of determination (R^2^) of 0.9974 indicates that the projected polynomial model fits the data tolerantly well. The predicted R^2^ (Pred R-Sq) value of 0.9771 agreed reasonably with the adjusted R^2^ (Adj R-Sq) value of 0.9942. Figure 4 shows a comparison of the predicted and actual percentages of MB removal. The predicted and actual values are in good agreement as the values are distributed along the regression line; furthermore, the values did not exceed the experimental range [47].

### 2.3. Response Surface Plots

Three-dimensional (3D) and two-dimensional (2D) plots were produced using Design Expert 7.0. The plots highlight the relationship between independent variables (*X_1_*, *X_2_*, and *X_3_*) and the response (MB removal), as presented in Figure 5, Figure 6 and Figure 7. For both contour and surface plots, two continuous variables were varied with MB removal as the response, while another variable was fixed at a zero level. Referring to Figure 5, the adsorbent dosage and MB concentration were varied, whereas the contact time was fixed at 2 h. The results clearly showed that the maximum MB removal obtained was 97.98% at approximately 1.5 g of adsorbent dosage and 200 ppm of dye concentration within 2 h. By increasing the adsorbent dosage, the adsorption efficiency increased significantly, as there was more available surface area and active sites and an increase of vacant sites for the adsorption of MB molecules on the adsorbent [48].

The behavior of MB removal, influenced by varied adsorbent dosages and contact times with the dye concentration fixed at 300 mg/L, is presented in Figure 6. The maximum MB adsorption (i.e., 96.07%) was obtained at a higher contact time. In the dye adsorption process by the adsorbent, first, the adsorbate would reach the boundary layer, before diffusing into the adsorbent surface, and later into the porous adsorbent structure, thus explaining the lengthy adsorption process [48]. The effect of different contact times and MB concentrations on MB removal is presented in Figure 7, with the adsorbent dosage set at a zero level. It could be observed that the percentage of MB removal increased as the MB concentration and contact time increased from 200 to 300 mg/L and from 1 to 2 h, respectively. Moreover, within those levels, the MB removal reached its maximum value.

To obtain the maximum MB elimination, the experiment was validated at the optimal settings, as seen in Table 3. Based on the initial parameters set to obtain the optimum response (*X_1_* = 1.5 g, *X_2_* = 200 mg/L, and *X_3_* = 2 h), the estimated MB removal was 95.5%, which was obtained at the optimum conditions of *X_1_* = 1.25 g, *X_2_* = 350 mg/L, and *X_3_* = 2.15 h, as shown in Table 4. An additional experiment was performed to validate the optimum conditions and the model agreement. From the results, the MB removal achieved was 97.75%, with a maximum error of only ± 2.3%. Hence, this small error reflects the validity of the response surface optimization conducted [49].

## 3. Materials and Methods

All chemical reagents were of analytical grade and used without further purification. Oil palm trunk chips were purchased from a local factory. Phosphoric acid (H_3_PO_4_, 80%) and MB were purchased from Sigma-Aldrich, (Petaling Jaya, Malaysia).

The surface morphology of oil palm trunk activated carbon (OPTAC) was characterized using a scanning electron microscope (model JSM-5910, JEOL USA, Peabody, MA, USA). A Nicolet iS20 spectrophotometer was used for the identification of chemical functional groups present in OPTAC through FTIR-ATR analysis. Meanwhile, a Bruker D2 Phase X-ray diffractometer with Cu-Kα (λ = 0.154060 Å) radiation source operating at 40 kV and 25 mA was utilized for studying the diffraction patterns of OPTAC. A UV-vis spectrophotometer (HACH DR6000) was used for the determination of MB removal percentage.

### 3.1. Preparation of OPTAC

First, water-soluble organic compounds were removed from OPT by rinsing the OPT several times with distilled water. Next, the sample was ground and sieved using a 0.5 mm sieve to obtain a uniform-sized sample followed by a pre-carbonization process for the sieved OPT in a furnace for 3 h at 350 °C. Subsequently, the sample was soaked in H_3_PO_4_ with a ratio of 1:10 (*w/w*) for 12 h and the mixture was stirred at 70 °C until the excess phosphoric acid was evaporated. The sample then was placed in an oven at 120 °C for an additional 12 h. Upon completion, the sample was heated in a furnace at 500 °C and 3 h for carbonization; subsequently, the sample was washed thoroughly and repeatedly using distilled water and basic solution to obtain a neutral pH. Lastly, the sample was inserted into an oven for drying at 120 °C for 12 h. The dried sample was kept in an airtight container prior to further use.

### 3.2. Dye Adsorption Study

In this study, MB was used as the adsorbate. The determined concentration of MB was placed in a 250-mL beaker and the prepared OPTAC was added to the beaker. Stirring was maintained at a stirring speed of 300 rpm and a temperature of 30 °C. A fixed volume of MB solution was collected at the determined contact time interval during each adsorption experiment to evaluate the amount of MB removed. The absorbance measurement of MB solution was conducted using a HACH DR6000 spectrophotometer at a wavelength of 662 nm. Equation (2) was used to calculate the MB removal percentage:(2)$$\mathrm{MB}\;\mathrm{Removal}\;\left(\%\right)=\frac{C_{o}-C_{i}}{C_{o}}$$
where *C_0_* and *C_i_* (mg/L) are the initial concentration and concentration at time t, respectively.

### 3.3. Design of Experiment

Three independent variables (adsorbent dosage, *X_1_* (g); MB concentration, *X_2_* (mg/L); and contact time, *X_3_* (h)) influencing MB removal from aqueous solution were evaluated by adopting a BBD, with MB removal percentage as the response. The actual and coded levels of independent variables used in the experimental design are presented in Table 3. The experimental design matrix was created using Design Expert 7.0 according to 33 factorial designs.

All experiments were carried out randomly to reduce the impact of systematic errors on the observed responses. For each component evaluated, a second-order polynomial equation was fitted as a function of *X*, as shown in Equation (3):(3)$$Y=\beta_{O}+\overset{3}{\underset{i=1}{\sum }}\beta_{i\;}X_{i}+\overset{3}{\underset{i=1}{\sum }}\beta_{ii\;}X_{i}^{2}+\overset{3}{\underset{\begin{array}{c} i=1\\ 1<j \end{array}}{\sum }}\beta_{ij}X_{i}X_{j}$$
where *Y* is the predicted response (MB removal, %); *β_0_*, *β_i_*, *β_ii_*, and *β_ij_* are the model regression coefficients; and *Xi* and *Xj* represent the independent variables (*i ≠ j*). The effect of each independent variable on the response was evaluated using the model, as well as the optimum response (*Y_opt_*) and corresponding independent variables.

Analysis of variance (ANOVA) was performed to check the model’s validity, where the analysis enables the determination of the relationship between the response and the process variables based on the proposed model. Meanwhile, the Fisher *F*-distribution test and *p*-value were used for evaluating the model’s statistical significance [50].

## 4. Conclusions

This research demonstrated that AC as an adsorbent was successfully synthesized from oil palm trunk (OPT). The characterization of the synthesized OPTAC using SEM revealed its porous structure. Meanwhile, the FTIR-ATR and XRD spectra showed a successful activation process and an amorphous phase of OPTAC were obtained, respectively. The optimization of methylene blue (MB) removal was carried out using a Box–Behnken design (BBD) of RSM for three independent variables (adsorbent dosage, MB concentration, and contact time). From the ANOVA results, an MB removal of up to 97.9% was obtained at the optimum conditions of *X_1_* = 1.5 g, *X_2_* = 200 mg/L, and *X_3_* = 2 h. Adsorbent dosage was the most influential parameter among the independent variables considered. Based on the verification experiment performed at the optimum conditions, the experimental MB removal (97.75%) closely agreed with the predicted MB removal (95.5%). Therefore, it can be concluded that the OPTAC prepared from agricultural waste is an excellent candidate as a low-cost adsorbent in the removal of contaminants from an aqueous solution.

## Figures and Tables

**Figure 1 molecules-26-04430-f001:**
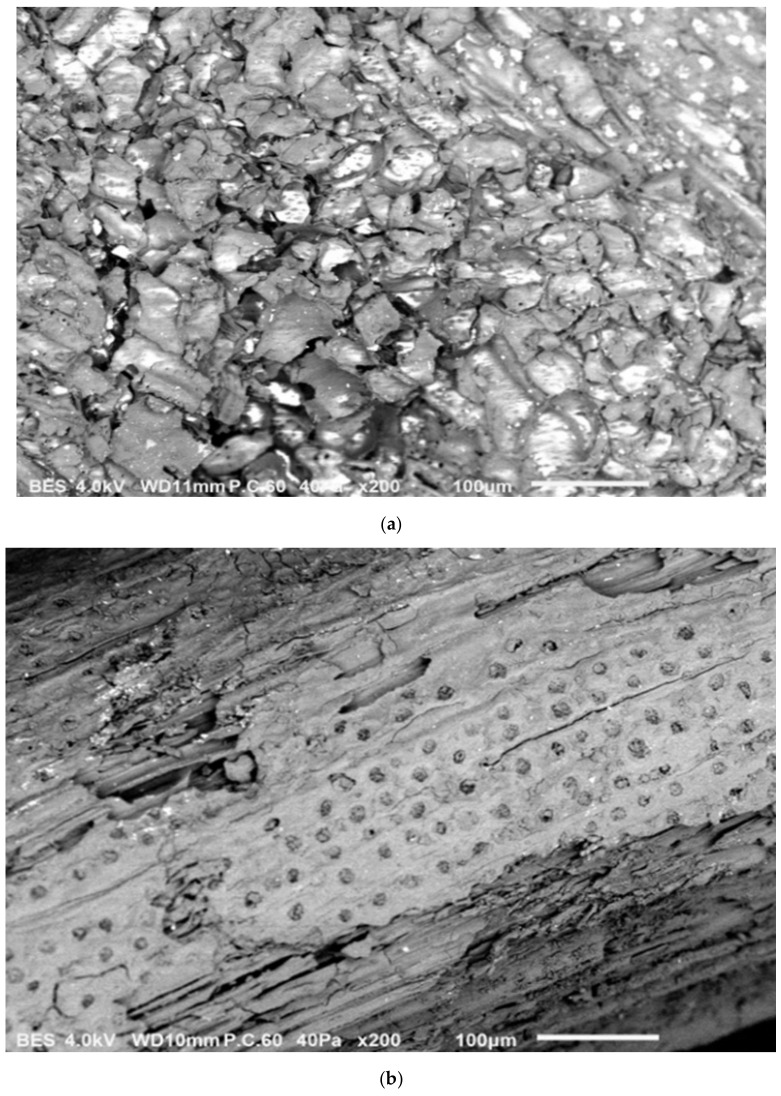
SEM images indicating carbon before activation (**a**), and after activation with H_3_PO_4_ (**b**), (**c**), and (**d**).

**Figure 2 molecules-26-04430-f002:**
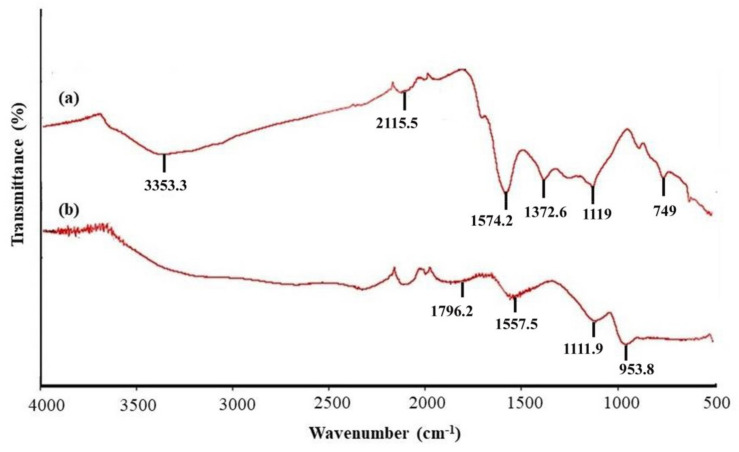
FTIR-ATR spectra for (**a**) NC and (**b**) OPTAC.

**Figure 3 molecules-26-04430-f003:**
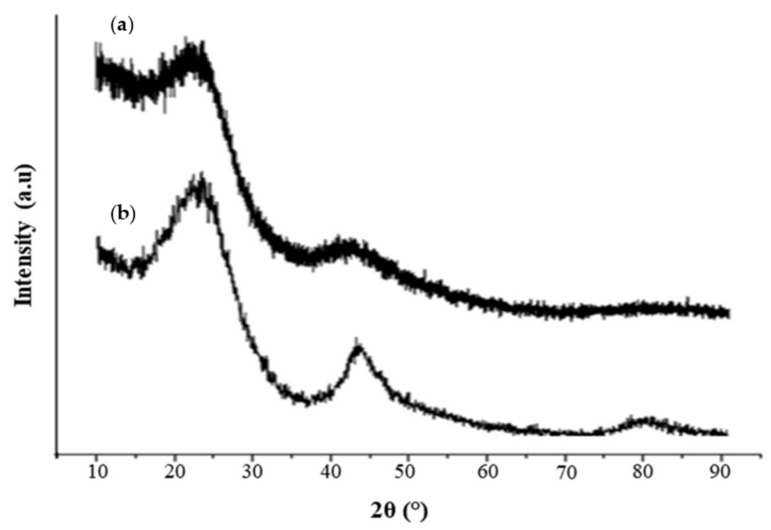
X-ray diffraction patterns of (**a**) NC and (**b**) OPTAC.

**Figure 4 molecules-26-04430-f004:**
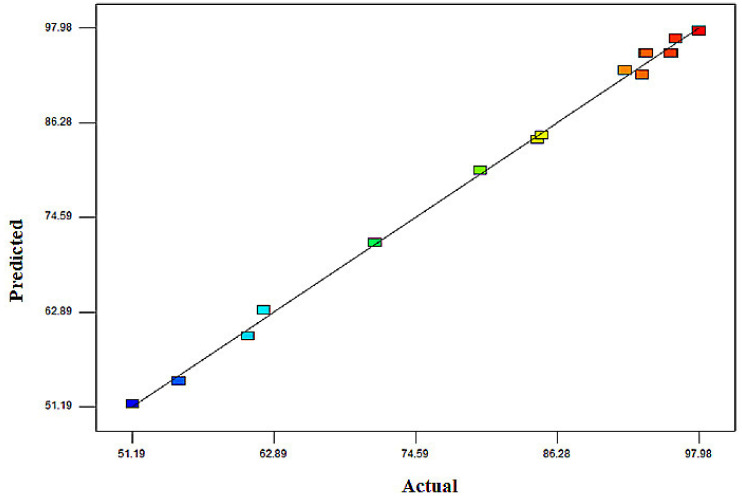
Correlation of predicted versus actual responses for the MB removal percentage.

**Figure 5 molecules-26-04430-f005:**
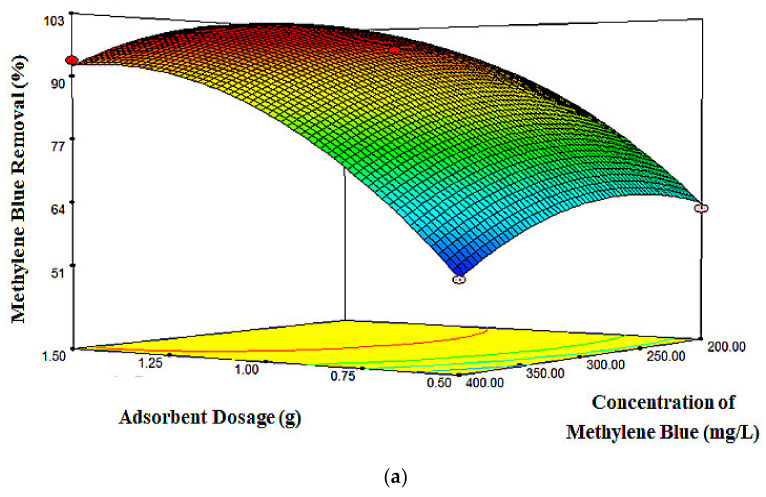
(**a**) 3D surface plot and (**b**) 2D contour plot for the effect of adsorbent dosage and MB concentration on MB removal percentage.

**Figure 6 molecules-26-04430-f006:**
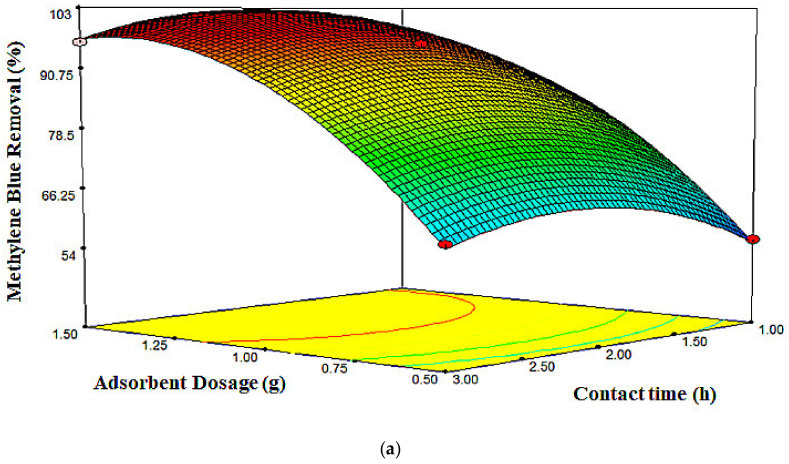
(**a**) 3D surface plot and (**b**) 2D contour plot for the effect of adsorbent dosage and contact time on MB removal percentage.

**Figure 7 molecules-26-04430-f007:**
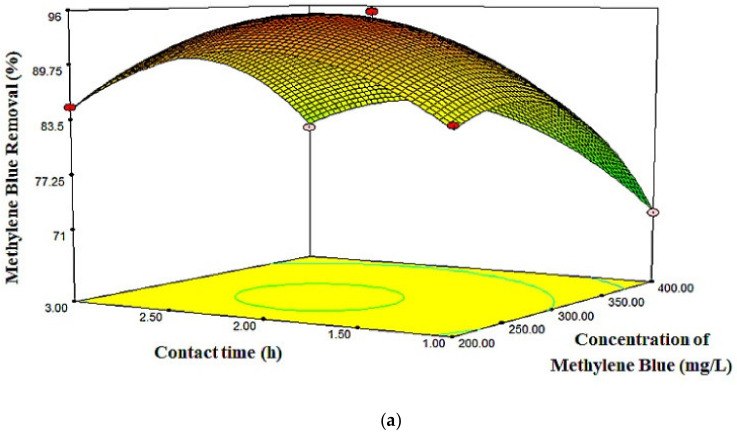
(**a**) 3D surface plot and (**b**) 2D contour plot for the effect of MB concentration and contact time on MB removal percentage.

**Table 1 molecules-26-04430-t001:** Experimental and predicted results for MB removal.

Run	*X_1_* (g)	*X_2_* (mg/L)	*X_3_* (h)	MB Removal (%)	
				Actual	Predicted
1	1	400	1	71.22	71.49
2	0.5	400	2	51.19	51.54
3	0.5	300	1	55.01	54.39
4	1	300	2	95.78	94.86
5	1	300	2	95.66	94.86
6	1.5	300	1	91.9	92.74
7	1	200	3	85.01	84.74
8	0.5	300	3	60.74	59.9
9	0.5	200	2	62.05	63.15
10	1.5	200	2	97.98	97.63
11	1.5	400	2	93.3	92.2
12	1	300	2	93.67	94.86
13	1	400	3	79.91	80.39
14	1	300	2	93.57	94.86
15	1	200	1	84.67	84.19
16	1.5	300	3	96.07	96.69
17	1	300	2	95.64	94.86

**Table 2 molecules-26-04430-t002:** ANOVA results for MB removal.

Source	Sum of Squares	Degree of Freedom	Mean Square	*F*-Value	*p*-Value
Model	4160.28	9	462.25	304.23	<0.0001
*X_1_*	2822.26	1	2822.26	1857.43	<0.0001
*X_2_*	145.27	1	145.27	95.60	<0.0001
*X_3_*	44.79	1	44.79	29.48	0.0010
*X_1_X_2_*	9.55	1	9.55	6.28	0.0406
*X_1_X_3_*	0.61	1	0.61	0.40	0.5470
*X_2_X_3_*	17.43	1	17.43	11.47	0.0116
*X_1_^2^*	557.16	1	557.16	366.69	<0.0001
*X_2_^2^*	220.14	1	220.14	144.88	<0.0001
*X_3_^2^*	232.49	1	232.49	153.0	<0.0001
Residual	10.64	7	1.52		
Lack of Fit	5.46	3	1.82	1.41	0.3636
Pure Error	5.17	4	1.29		
Std. Dev.	1.23	16	R^2^	0.9974	
Mean	82.55		Adj R^2^	0.9942	
			Predicted R^2^	0.9771	

**Table 3 molecules-26-04430-t003:** Experimental ranges and levels of independent variables.

Independent Variables	Range and Level		
	−1	0	1
Adsorbent Dosage, *X*_1_ (g)	0.5	1	1.5
MB Concentration, *X*_2_ (mg/L)	200	300	400
Contact Time, *X*_3_ (h)	1	2	3

**Table 4 molecules-26-04430-t004:** Values of process parameters for the maximum MB removal percentage.

Parameters	Values
Percentage of MB Removal (%)	95.50
*X_1_* (Adsorbent Dosage, g)	1.25
*X_2_* (MB Concentration, mg/L)	350
*X_3_* (Contact Time, h)	2.15

## Data Availability

All data is contained within the article.

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
