# Peer review of "Enhancing the Decolorization of Methylene Blue Using a Low-Cost Super-Absorbent Aided by Response Surface Methodology"

_molecules, 2021, doi:10.3390/molecules26154430_

Round 1
Reviewer 1 Report
Experimental design is a useful method to complete a deep adsorption study. However, the investigations about the MB adsorption onto the AC are poor.
First, AC obtention from different biomass as well as different waste are well known. In addition, H3PO4 activation is also widely investigated by several authors. Thus, what is new in the AC obtention?
Different techniques were used for the characterization of the OPTAC such as FTIR and XRD. However, N2 adsorption-desorption isotherm which is a common technique used to the characterization of these types of materials could improve the work. Parameters that are calculated used these measurements are directly related to the adsorption onto the AC.
On the other hand, MB adsorption is also a well-known dye, and this adsorbent process is previously described. So, ANOVA analysis is right, but the typical kinetic and isothermal analyses are lost. Regarding adsorption experiments, why these conditions were used by the authors? A Larger contact time and higher adsorbent amount could not lead an efficient adsorption process. In addition, the parameters analyzed lead to expected results. An increase of the adsorbent dosage, as well as low MB concentration, promotes an improvement of the adsorption. Other parameters which affect the adsorption process such as temperature, pH solution, ionic strength... were studied by the authors?
So, the manuscript needs a high improve to be published in Molecules journal, and it can not be accepted for publication in the present form.
Reviewer 2 Report
The analysis of adsorption behavior by the response surface method is a unique approach. However, I could not be convinced to the appropriateness of the parameter examined. Contact time is a suitable parameter, while concentration of dye seems not a parameter for optimization. Practical polluted liquids contain pollutants with a wide range of concentration. Adjusting the amount of the adsorbent depending on the amount of pollutants may be cost-effective, but needs a process for determination of the concentration. I think that a similar analysis on the other parameters like pH and the synthetic conditions of the adsorbent would give useful information.
I recommend changing the aim of this work to enhancing the cost and ecological effectiveness. The use of the minimum amount of the adsorbent for pollutants is economically and ecologically (to reduce the amounts of adsorbents, not for complete removal of pollutants) desirable. The conditions optimized in this work seem not enhance the efficiency itself. Use of excess amounts of adsorbents is desirable for complete removal.
Experiments
The approximate size of the ground OPT should be indicated.
Was OPT after soaking in phosphoric acid rinsed or wiped? The amount of phosphoric acid before carbonation would affect the morphology of OPTAC.
Result and discussion
The synthesis of OPTAC (not the procedure explained in the experimental section) should be described. For example, the appearance of OPT after soaking is informative (swelling with phosphoric acid, change in color and morphology). Of course, the change in the morphology (more macroscopic one than SEM images) and weight by carbonation are also informative.
The SEM images are relatively confusing. I could not understand the typical morphology of the obtained carbon materials with the different images with the same magnification for one material. Addition of the images with lower magnification helps the readers to catch the whole aspects.
The assignment of the peaks in the FTIR spectra is uncertain, while these peaks are assignable to the proposed structures. The assignment should be indicated as possible assignments.
Others
Many grammatical errors were found. Careful corrections are necessary. In addition, the use of multiple names for a material with some mistakes (e.g., OPTAC was referred as oil palm trunk at the bottom of page3, and called with other names in the other parts) is very confusing. The abbreviations (MB, OPTAC, NC) should be used throughout the manuscript after the first abbreviation.
Round 2
Reviewer 1 Report
The manuscript can be accepted for publication
Author Response
Thank you for time and valuable comment. We really appreciate that.

Reviewer 2 Report
Comment on your replies 1 &2
My point is that the examined parameters are not to optimize the conditions, but are to know the ability of the adsorbent. How do you apply these parameters to practical solutions? Do you measure the concentration of pollutants before purification and calculate the appropriate amount of the adsorbent to be used? I think this is not a typical procedure. I accordingly recommended changing the aim of the work to maximize the cost effectiveness or adding other parameters of conditions.
Comment on your reply 3.2
OPT was soaked in phosphoric acid. I presume that the soaked one was removed from phosphoric acid by any procedure. The soaked OPT should contain phosphoric acid, and its amount is important (if possible, the amount of phosphoric acid contained should be indicated). Accordingly, the removal process (e.g., filtration, vacuum filtration, or centrifugation and rinsing) before carbonation must be indicated. If the suspension of OPT in phosphoric acid was directly heated, the ratio of phosphoric acid and OPT must be indicated.
Comment on your reply 4.1
Usually, new experiments are explained in the result and discussion section, while results according to known ones are not. If OPTAC in this work is new, I regard that the aspect of the synthesis should be explained. Experimental section is to explain only the method and data. Such styles of writing may depend on authors and journals, and I leave the judgement to the editor.
Comment on your replies 4.2 and 4.3
- The figure and the explanation became very understandable.
However, the description on the assignment of the FTIR spectra is too strong. These peaks suggest the presence of these chemical structures, but are not the certain proofs. The use of words indicating the possibility (e.g., assignable, suggesting the presence of, agreeable, plausibly) should be used.
Some inadequate expressions were corrected, but appropriate editing is still necessary.
